# Agronomic and Economic Aspects of Biodiesel Production from Oilseeds: A Case Study in Russia, Middle Volga Region

Kirill A. Zhichkin [1], Vladimir V. Nosov [2,3,4], Lyudmila N. Zhichkina [5], Elena A. Krasil'nikova [3], Olga K. Kotar [6,*], Yuri D. Shlenov [4], Galina V. Korneva [4], Anna A. Terekhova [4], Vadim G. Plyushchikov [2], Vladimir P. Avdotin [2], Regina R. Gurina [2] and Tatiana V. Magdeeva [2]

1   Department of Economic Theory and Economics of the Agro-Industrial Complex, Samara State Agrarian University, 446442 Kinel, Russia
2   Technosphere Security Department, Peoples' Friendship University of Russia (RUDN University), 117198 Moscow, Russia
3   The Basic Department of Trade Policy, Plekhanov Russian University of Economics, 117997 Moscow, Russia
4   Department of Finance, Accounting and Economic Security, K.G. Razumovsky Moscow State University of Technologies and Management, 109004 Moscow, Russia
5   Department of Agrochemistry, Soil Science and Agroecology, Samara State Agrarian University, 446442 Kinel, Russia
6   Department of Accounting and Statistic, Saratov State University of Genetics, Biotechnology and Engineering Named after N.I. Vavilov, 410012 Saratov, Russia
*   Correspondence: kotarok@internet.ru

**Abstract:** Emissions from fossil fuels are expected to increase in accordance with the global economy, which causes the development of alternative non-hydrocarbon sources in energy production. Biodiesel is one of the best options, among other sources, due to its low footprint. Russia does not have a smart policy of state support for biofuel production. The work objective was to determine whether it is necessary to develop equipment for biodiesel production, taking into account the structure of cultivated areas and available technologies; to calculate economic indicators of biodiesel production for agricultural needs; to compare the options for spring rape cultivation; as well as calculate the government support optimal level. As research methods, the authors used the apparatus of economic and mathematical modeling, and the method of absolute, relative and average values. Statistical tables are used to present the research results. Based on our study results, it is proven that the homemade biodiesel production by agricultural enterprises is economically justified. The equipment needed for its production was determined, the main economic indicators of the fuel production type and the optimal value of monetary and labor costs were calculated, and the gross and market biofuel values were obtained. The optimum level of government support for biofuel production in the Middle Volga region should be EUR 13.223 million, and the area planted with oil crops should be increased by 47.1 thousand ha.

**Keywords:** biodiesel; spring rape; oilseed cultivation; sunflower; the Samara region; methyl ester

## 1. Introduction

By 2030, worldwide energy consumption is expected to increase by 53%, and greenhouse gas emissions from fossil fuels by 39% [1–5]. Currently, however, increasing attention is paid to the use of alternative fuels, due to the reduction in the worldwide supply of geogenic energy carriers, tightened exhaust emission standards, and limitation of carbon monoxide emission [6–10]. As an alternative fuel, biodiesel is one of the best options, among other sources, due to its environmental friendliness and functional properties similar to diesel fuel [11]. Governments promote the development and use of biofuel. For example, in the USA, in accordance with the adopted program, the share of renewable fuels has increased by 10% over the period of 2005 to 2017 [12–14]. In member states of the European

Union, a directive on the promotion of the use of biofuels was adopted, under which it was required to achieve a contribution of at least 10% of biofuel in total fuel consumption by 2020 [15]. Biofuel production is also considered an important strategy to achieve the goals of the Paris Agreement [16]. Despite Russia being one of the largest oil exporters, many Russian scientific and manufacturing institutions have taken an active interest in the production and consumption of environmentally friendly bioenergy carriers produced from renewable biological feedstock [17–19].

Its chemical composition allows it to be used in diesel engines without the substances that stimulate the ignition of biodiesel. The following useful properties of biodiesel should also be noted:

- biodiesel undergoes almost complete biological decay: in the soil or in water, microorganisms recycle 99% of biodiesel in 28 days;
- less $CO_2$ emissions;
- low number of components in exhaust gases, such as carbon monoxide (CO), unburned hydrocarbons and soot;
- low sulfur content;
- good lubricating characteristics. An increase in the service life of the engine and fuel pump by an average of 60% must be achieved [20,21].

In order to calculate the possibility of developing biofuel production in the Samara region and providing the region's agriculture with biofuel (biodiesel), it is necessary to choose a crop, determine the most profitable cultivation technology, select equipment (identify the number of complexes necessary to provide the region's agriculture with biofuel and calculate its payback), and consider the possibility of state support for this project. It is important to know how different technologies for the cultivation of oilseeds will affect the cost of the produced crop products and other resulting technical and economic indicators of the technology. For this purpose, an economic assessment of various technologies for the cultivation of oilseeds, used in the arable fields of the Samara region, was carried out on the example of the cultivation of spring rape. A feature of spring rape as a crop is the possibility of using rape as green manure fallow, and the grower can cultivate cash crops in the saved areas [22–25]. In the conditions of the Samara region, the cultivation of spring rapeseed for the production of biodiesel is the most appropriate. Winter rapeseed in this region has not yet been adapted and cannot overwinter. Other crops are characterized by either low yields or high manual labor costs (pumpkin). Sunflower has a high market potential as a basis for the production of sunflower oil for food purposes. Biodiesel cannot compete with it.

The use of biodiesel in agricultural production can reduce the cost of manufactured goods and increasing production efficiency. Government support for the development of biofuel production is a relevant part of the system of budget regulation of agricultural production. Currently, there is no sound financing method for mechanisms of state regulation of biofuel production, which impedes impartial allocation of funds and makes this procedure non-transparent and not sufficiently appealing.

In addition, in the production of biofuels, the Russian Federation will have the opportunity to ensure the sale of agricultural products, produce more than 7.5 million tons of biodiesel, and increase agricultural productivity. This will improve the environment and the health of citizens, and increase the income of agricultural producers. Moreover, the import field for trading in greenhouse gas emissions under the Kyoto Protocol will expand. An additional plus is the lower cost compared to the purchase price of diesel fuel.

## 2. Materials and Methods

Agronomic, technical and economic information necessary for the effective implementation of crop production technology requires a technological map. For this calculation, the methodology for developing technological maps was used in accordance with national standard 10 1.3 for each variant of the technology of rapeseed cultivation.

To solve the problem posed in the present study, specialized software for calculating technological maps in crop production, developed in 2006 by scientists of Samara State Agrarian University, was used (Figure 1). This software allows the user to calculate the production costs based on the list of technological operations, the set of equipment used, fertilizers, seeds, plant protection products and other factors. It is a database of technological operations adapted to specific climatic, soil conditions, cost characteristics and technologies. With its help, the diesel fuel amount needed to complete the production program was determined [26].

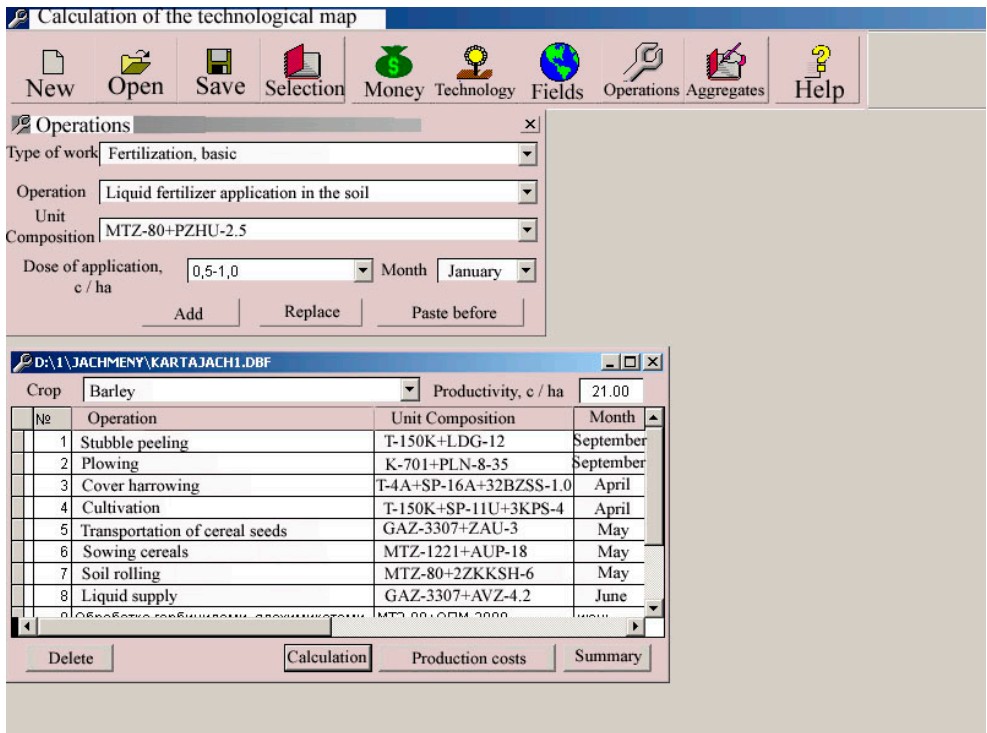

**Figure 1.** An example of filling in the initial information for calculating the technological map in crop production.

In order to achieve the purpose of the study, we used statistical data provided by agricultural organizations of the Samara region (Figure 2), disaggregated by economic zones (Northern, Central and Southern) that differ in technologies used and varieties of crops. On 01/01/2020, 1001 agricultural production units and organizations were registered. In the Samara region, agricultural area accounts for 3799.8 thousand ha of agricultural land, including 2871.2 thousand ha of arable land (75.6%), 95.1 thousand ha of fallow (2.5%), 27.5 thousand ha of perennial planting (0.7%), 50.7 thousand ha of hayfields (1.3%) and 755.3 thousand ha of pasture (19.9%).

Considering that the total area of arable lands in the Samara region is 2871.2 thousand ha, and that, annually, agricultural production consumes 60 kg of diesel fuel per ha of arable land, the average consumption of diesel fuel is 172.2 thousand tons per year.

Subsequently, the monographic, abstract logical method, situational and system analysis, economic statistical methods, and the method of expert evaluations were used. The results of the research are presented in tabular and graphical forms.

The present study suggests the development of a mathematical model for the optimization of government support for biodiesel production. This economic mathematical model is based on principles of linear programming [27–30].

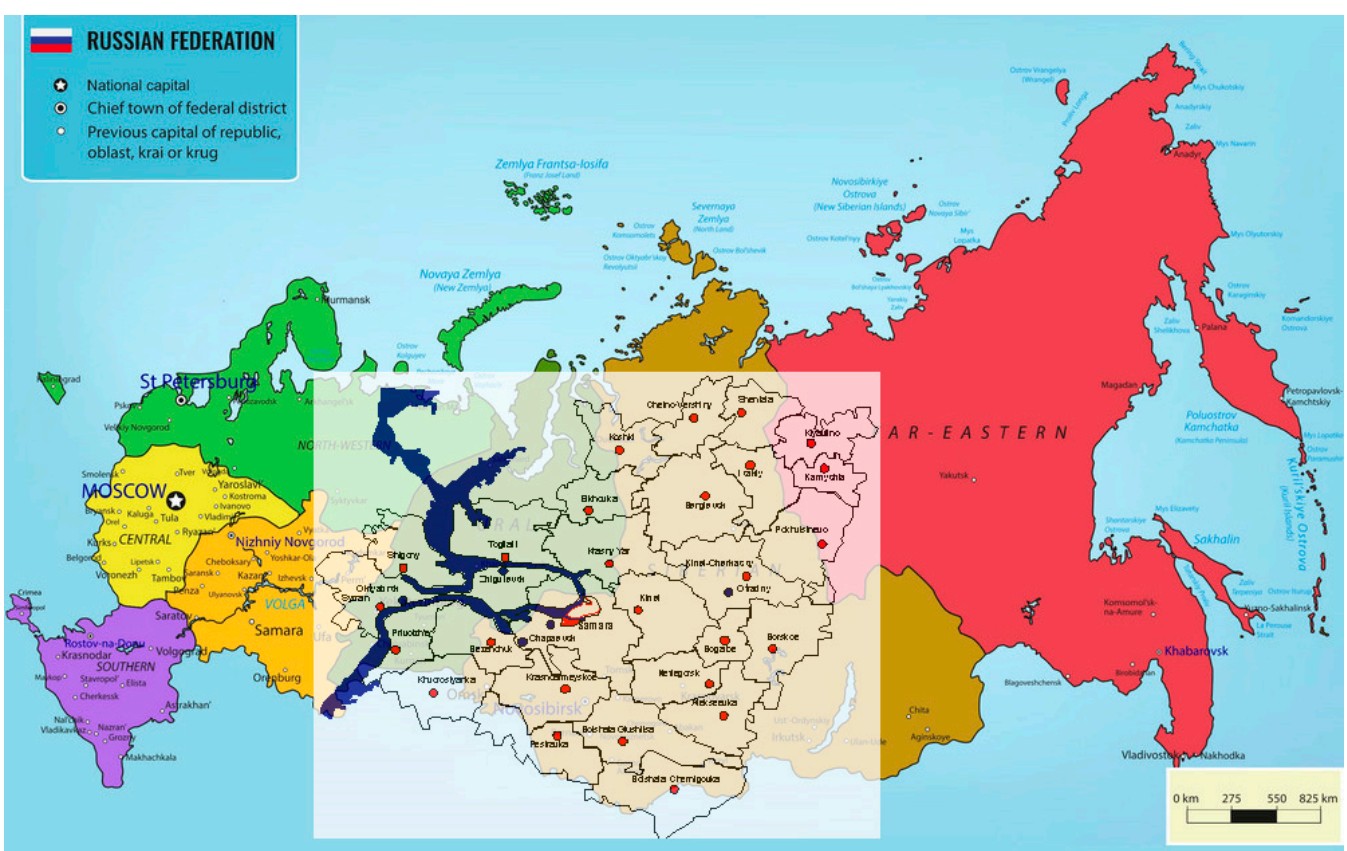

**Figure 2.** The Middle Volga region (the Samara region).

The mathematical model includes the following components.

The objective function, subject to maximization or minimization:

$$Z_{(\max,\min)} = \sum_{j=1}^{n} c_j x_j = c_1 x_1 + c_2 x_2 + \ldots + c_n x_n \tag{1}$$

where $n$ is the total number of unknown variables; $j$ is the sequential number of a variable ($j = 1, 2, \ldots, n \in N$); $c_j$ is the evaluation of the objective function per $j$ unit; $x_j$ is unknown.

A system of linear inequalities:

$$
\begin{cases}
a_{11}x_1 + a_{12}x_2 + \ldots + a_{1j}x_j + \ldots + a_{1n}x_n \le (=, \ge)b_1 \\
a_{21}x_1 + a_{22}x_2 + \ldots + a_{2j}x_j + \ldots + a_{2n}x_n \le (=, \ge)b_2 \\
\ldots\ldots\ldots\ldots\ldots\ldots\ldots\ldots\ldots\ldots\ldots\ldots\ldots\ldots\ldots\ldots\ldots\ldots\ldots\ldots\ldots\ldots\ldots \\
a_{i1}x_1 + a_{i2}x_2 + \ldots + a_{ij}x_j + \ldots + a_{in}x_n \le (=, \ge)b_i \\
\ldots\ldots\ldots\ldots\ldots\ldots\ldots\ldots\ldots\ldots\ldots\ldots\ldots\ldots\ldots\ldots\ldots\ldots\ldots\ldots\ldots\ldots\ldots \\
a_{m1}x_1 + a_{m2}x_2 + \ldots + a_{mj}x_j + \ldots + a_{mn}x_n \le (=, \ge)b_m
\end{cases}
\tag{2}
$$

where $a_{ij}$, $b_i$ are given constants; $i$ is the sequential number of constraints ($i = 1, 2, \ldots, m$).

Non-negativity constraints on all variables included in the system:

$$x_j \ge 0 \tag{3}$$

The model should have the ability to apply to individual households and household groups of different forms of business organization.

## 3. Results and Discussion

Fatty oils and, less often, essential oils obtained from plants and algae, are used as raw materials for biofuel production. Used cooking oil, animal fats and fish oil are also used in production [31–35]. Note that biofuel produced from specific oils has different properties [36–38]. Table 1 shows the crops cultivated in the Samara region that can be used in biofuel production.

**Table 1.** Production of oil from crops per ha.

| Crop | Kilograms of Oil per ha (kg/ha) | Liters of Oil per ha (L/ha) |
|---|---|---|
| Corn | 145 | 172 |
| Oat | 183 | 217 |
| Lupine | 195 | 232 |
| Soybean | 375 | 446 |
| Flax seeds | 402 | 478 |
| Pumpkin seeds | 449 | 534 |
| Mustard seeds | 481 | 572 |
| Milk-cap | 490 | 583 |
| Sunflower | 800 | 952 |
| Rapeseed | 1000 | 1190 |

Rapeseed oil is oxidation-resistant and has an iodine number of less than 120; therefore, rapeseed oil is convenient to use in winter. Rapeseed produces high yields, hence why most areas are planted with this crop, which is later used for biofuel production.

Sunflower oil is also used for biofuel production. Currently, sunflower yields are lower than rapeseed yields, but it grows well in countries with a warm, dry climate. It has an iodine number of more than 120 (according to the European Standard EN 14,214, it should not exceed 120), which is why it needs to be blended with other oils that contain less iodine. Potential possibilities of using other oilseeds as raw materials for biofuel production have not been fully explored yet [31].

For the production of biofuel with developed properties, the following crops can be used:

- crops with minimum concentrations of polyunsaturated fatty acids, such as linoleic acid (18:3);
- crops with maximum concentrations of monounsaturated fatty acids, such as oleic acid (18:1), to ensure stability in combination with convenience of use in winter;
- crops with minimum concentrations of saturated fatty acids (16:0) and stearic acid (18:0) for convenient usage in winter [39–41].

Several important properties of products of transesterification of the most common vegetable oils with methanol are presented in Table 2.

**Table 2.** Properties of biofuel manufactured from different raw materials.

| Oil | Tpl, °C | Cetane Number |
|---|---|---|
| Rapeseed or soybean | −10 | 55–58 |
| Sunflower | −12 | 52 |
| Corn | −10 | 53 |

Currently, the most common biofuel is rapeseed methyl ester (RME), which is extensively used in Sweden, Germany, France and other countries. Up to 30% of it can be added to diesel fuel without additional engine modification. Western European countries have decided on a mandatory addition of 5% of RME to diesel fuel, though in some countries (Sweden, for example), RME is used as a substitute for diesel. Thus, we propose that the production volume of methylated vegetable oils will increase and agri-technologies will improve, which will result in the reduction in their costs to an acceptable level [42–44].

Many scientific research institutions and universities, including Samara State Agrarian University and the Povolzsky machinery testing station, have conducted research on the use of RME biofuel, and developed utility flow schemes as well as fuel supply systems for tractors, adapted for the use of biofuel.

It was established that the reduction in engine power output of biofuel is insignificant: fuel consumption increases by 5–8%, but engine life does not change. Biofuel also has promising lubricating properties. Soot emissions decrease by 50%, carbon dioxide by 10–12% and sulfur by 0.05%, as compared with 0.2–0.5% for diesel fuel.

Technology for the conversion of vegetable oils to biofuel has developed considerably over the past few years, especially in Tatarstan. The resulting products (diesel fuel, forage pulp and glycerin) are in demand, and their joint production makes the process cost-effective.

The simplicity of the technology and economic characteristics of the process makes biofuel more appealing to agricultural producers, considering that diesel fuel is the main fuel in agriculture.

The first organization to produce biofuel in the Samara region was "Biosam" in Krasnoyarsky district. On the basis of the laboratory of the Department of Tractors and Vehicles of Samara State Agrarian University, the Biosam company tested biofuel samples produced by MIXER. The samples demonstrated great antiwear and antiscuffing results.

The results of bench tests conducted on engines running on alternative fuel conducted by the Povolzsky machinery testing station show that:

- engine power generated by blends of diesel and biofuel in different proportions is sufficiently close to the engine power generated by diesel fuel, is within tolerance limits, and differences are insignificant. A slight increase in engine power for 50% biofuel blend is due to high kinematic viscosity of blends, which reduces leakage in plunger pairs;
- fuel consumption rate for engines running on diesel–biofuel blend is higher than for diesel fuel due to the lower calorific value of biofuel.

We also calculated the comparative effectiveness of biofuel production. In accordance with this calculation, the cost of 1 L of homemade biofuel is 30–50% lower than the wholesale price of diesel fuel.

To determine the minimum number of biofuel production units, we should determine the amount of fuel consumed in the Samara region, which depends on cultivated crops and production technology. Table 3 shows the structure of crop acreage by agricultural zones in percentage.

**Table 3.** Structure of crop acreage by agricultural zones of the Samara region (%).

| Area | Total Crop Acreage | Grain and Leguminous Crops | Industrial Crops | Potato and Cucurbits | Forage Crops |
|---|---|---|---|---|---|
| Northern zone | 25 | 57.12 | 26.61 | 0.05 | 16.22 |
| Central zone | 32 | 53.95 | 28 | 0.58 | 17.47 |
| Southern zone | 43 | 56.4 | 34.13 | 0.64 | 8.83 |
| Region | 100 | 55.81 | 30.29 | 0.47 | 13.43 |

Annual reports of organizations reflect crop acreage and material costs of petroleum products per crop. Thus, material costs of petroleum products per ha can be calculated by dividing the material costs of petroleum products by crop acreage. Table 4 reports the results for agricultural zones of the region.

Considering that 15 organizations have not planted any cropland with potato and cucurbits, and that the share of this category in the structure of crop acreage does not exceed 0.5%, we decided to apply the average value for all categories to this category. To calculate material costs of petroleum products for every zone, we multiply the coefficients for crop categories by crop acreage.

**Table 4.** Material costs of petroleum products for cultivation of agricultural crops in zones.

| Crops | Material Costs of Petroleum Products per ha, Thousand EUR | | |
|---|---|---|---|
| | Northern Zone | Central Zone | Southern Zone |
| Grain and leguminous crops—total | 1.49 | 1.19 | 1.79 |
| Grain and leguminous crops (winter and spring) excluding corn including: | 1.66 | 1.19 | 1.71 |
| winter grains | 1.93 | 1.36 | 1.99 |
| spring grains | 1.34 | 1.18 | 1.34 |
| grain legumes | 1.24 | - | 1.72 |
| Grain corn | 1.30 | 1.03 | 2.20 |
| Industrial crops—total | 1.84 | 3.58 | 1.89 |
| Soybean | 1.66 | | - |
| Milk-cap | - | - | 1.68 |
| Common flax (cultivation) | - | - | - |
| Grain sunflower | 2.02 | 3.58 | 2.11 |
| Potato and cucurbits—total | 1.30 | 1.76 | 1.34 |
| Forage crops—total | 0.83 | 0.76 | 0.66 |
| Perennial grass | 1.06 | 0.30 | 0.34 |
| Annual grass | 0.53 | 0.88 | 0.69 |
| Silage corn and green corn | 0.90 | 1.09 | 0.95 |

Thus, EUR 28,460.73 thousand was spent for the cultivation of crops on the total area of 1,488,898.2 ha in the Samara region (Table 5). We analyzed trends in diesel retail prices using the Yandex Quotation, and the results show that, in 2018, the retail value of diesel fuel varied from EUR 0.367 to 0.382 per liter. However, considering the fact that agricultural units purchase large consignments of diesel fuel before field works, we can assume that the cost of diesel fuel is EUR 0.37/L.

**Table 5.** Material costs of petroleum products from cultivation of agricultural crops in areas of the Samara region in 2018.

| Region | Total Material Costs of Petroleum Products, Thousand EUR | Grain and Leguminous Crops | Industrial Crops | Potato and Cucurbits | Forage Crops |
|---|---|---|---|---|---|
| Samara | 28,460.72 | 14,453.90 | 12,170.57 | 127.56 | 1708.66 |

Therefore, we divide material costs by cost of fuel purchase. Note that material costs of petroleum products relate to costs of diesel fuel and gasoline. The structural analysis of petroleum products used over the past 3 years shows that diesel fuel accounts for 93%, and gasoline for 7%. Given the correction, we can determine the amount of petroleum products used to cultivate crops (Table 6).

**Table 6.** Petroleum products used in cultivation of agricultural crops by areas of the Samara region in 2018.

| Region | Petroleum Products, Tons | Grain and Leguminous Crops | Industrial Crops | Potato and Cucurbits | Forage Crops |
|---|---|---|---|---|---|
| Samara | 77,062.9 | 39,136.8 | 32,954.2 | 345.4 | 4626.7 |

Biodiesel is a methyl ester. It is obtained from vegetable oils by a transesterification reaction: methanol is added to the vegetable oil in a ratio of approximately 9:1 and a small amount of catalyst. From one ton of vegetable oil and 111 kg of alcohol (in the presence of 12 kg of catalyst), approximately 970 kg (1100 L) of biodiesel and 153 kg of primary glycerol are obtained. It is recommended to use potassium or sodium methoxide (methylates) as catalysts, after which the mixture is processed in a cavitation reactor [45,46].

Biosam LLC (Samara) developed an automated system, MIXER, for the production of biodiesel with a capacity of 500 and 1000 L per hour. The power consumption of cavitation reactors is 7.5 and 15 kW, respectively. The operating mode is three-shift. The complexes are equipped with metering devices that allow feeding components into the reactor with high accuracy. The use of a hydrodynamic cavitation reactor allows one to reduce the reaction temperature to 30–35 °C, and to ensure that all components are fully involved in the transesterification reaction [47,48].

After processing in a cavitation reactor, the mixture is fed into special separator columns, where it is divided into fractions. The complexes can work on absolutely any kind of vegetable oil, and use methyl alcohol, potassium hydroxide, sodium hydroxide or acid as catalysts in the reaction.

A flexible dosing system allows one to configure the installation not only with existing technologies for mixing the starting components, but also to create new technology that will most accurately take into account the local characteristics of the raw material [49].

The calculation was carried out for a semiautomatic device with a 6YL-80 press with a capacity of 600 kg/change of finished fuel. The biofuel thus obtained has passed the test in Samara State Agrarian University and the Povolzsky machinery testing station, which shows that its characteristics correspond to the fuel developed in the traditional way.

The cost of 1 kg of biodiesel from rapeseed oil will be EUR 0.308. (Table 7) According to the Ministry of Agriculture of the Russian Federation, the average price in Russia for summer diesel fuel as of 1 August 2018 amounted to EUR 0.421/ton.

**Table 7.** Formation of the cost of biodiesel from rapeseed oil.

| № | Indicator | Natural Indicator | Cost, Thousand EUR |
|---|---|---|---|
| 1 | Rapeseed yield | 1.5 t/ha | |
| 2 | The cost of capital investments (reconstruction of premises and purchase of technological equipment) | | 29.20 |
| 3 | Cost of components, total | 1 t | 0.18 |
| | including rapeseed oil | 885 kg | 0.16 |
| | methanol | 100 kg | 0.01 |
| | potassium hydroxide | 15 kg | 0.01 |
| 4 | Depreciation | | 2.92 |
| 5 | Salary | 2 men | 5.25 |
| 6 | Electric power | 11,088 KWh | 0.62 |
| 7 | Repair and maintenance costs | | 0.73 |
| 8 | General running costs | | 13.33 |
| 9 | Total production costs per year | | 66.63 |
| 10 | The cost of rapeseed oil biodiesel | 1 t | 0.31 |
| | For comparison (as of 1 August 2018): | | |
| 11 | The price of summer diesel fuel average for Russia | 1 t | 0.42 |
| 12 | Gas station price | 1000 l | more 0.40 |

The next task was to determine the comparative effectiveness of the sale of sunflower oil and the production of biodiesel from rapeseed oil [50–55]. A comparative calculation is carried out on the basis of a conventional farm with an area of arable land of 1000 ha.

With an average diesel fuel demand of 60 kg/ha, arable land is necessary for the entire agricultural complex to carry out 60 tons of fuel per year.

Option is sunflower.

In accordance with agricultural requirements, sunflower can be sown on the same area every 8 years.

This means that the maximum possible sowing area is 125 ha.

Sunflower yield will be 1.0 t/ha.

In total, one can obtain 125.0 tons of sunflower.

The yield of oil from sunflower seeds is 45%.

In total, 56.25 tons of oil can be obtained.

The cost of transport services and processing is EUR 1.065 thousand.

At a selling price of EUR 0.642/kg (data from the Ministry of Agriculture of the Russian Federation as of 01/08/2018), oil revenue will be EUR 36.095 thousand.

With these funds, one can purchase 85.8 tons of diesel fuel of the L-0.2-62 brand (summer) produced by the Novokuybyshevsky Oil Refinery (specific gravity of 0.86 kg/L).

To acquire 60 tons of diesel fuel, 87.4 hectares of sunflower must be sown.

At a cost of EUR 153.298/ha, total cost will amount to EUR 13.398 thousand.

Option is rapeseed.

With a yield of 1.5 t/ha and an area of 125 ha, 187.5 t of rapeseed can be obtained.

The yield of oil is 42% or 78.75 tons.

From this amount of oil, 87.81 tons of biofuel can be prepared.

Based on the needs of the economy, 85.42 ha can be allocated for rapeseed.

The cost of 1 ha of rapeseed will amount to EUR 115.047.

The cost of components is EUR 1.551 thousand.

The cost of cultivation is EUR 10.102 thousand.

Total cost—EUR 11.653 thousand.

The direct economic effect of replacing sunflower with rapeseed will be EUR 1.745 thousand/1000 ha.

Moreover, there is an indirect effect, expressed as follows:

- excess power churn can be used for squeezing oil for other purposes;
- improvement of soil fertility and phytosanitary situation;
- obtaining an additional amount (29.3 tons) of high-protein feed for dairy cattle (rapeseed cake). Rapeseed cake has a higher feed value (protein content up to 40% versus 32–34% in sunflower). Currently, the sale price of oilcake is EUR 125–150, i.e., the farm may receive additional income in the amount of EUR 3.66 thousand.

In conclusion, the comparative efficiency of biodiesel and diesel fuel for a particular economy was calculated.

State policy aimed at supporting the interests of the fuel and energy complex forces farmers to seek new approaches to agricultural production. Among those tested with identified locations, we can name the use of minimal and zero tillage, the use of energy-saving machines, etc. Currently, we are talking about finding a replacement for traditional fuel, i.e., the use of renewable energy sources, including vegetable oil based biodiesel. To determine the efficiency of using biodiesel, using the program for calculating technological maps in crop production, the need for diesel fuel was determined by month of the year.

The greatest demand falls on the month of October, when plowing is carried out, which is the most energy-intensive operation in traditional crop cultivation technology. To determine comparative efficiency, it is proposed that one uses the sowing structure of 2018, since, at present, it is this structure that most accurately reflects the financial and organizational capabilities of the enterprise in the field work [56–60]. It is proposed that one should add to this structure the necessary area for the cultivation of rapeseed (400 ha), which is sufficient to provide biodiesel for the economy. As a result, the sown area will be 3952 ha. Based on the per hectare demand for diesel fuel for the cultivation of these crops, as well as the size of the sown area, we determine the cost of diesel fuel in the current year. The total demand for diesel fuel will be 238.8 tons. To determine the cost of diesel fuel, the amount found must be multiplied by the purchase price of fuel and lubricants (EUR 0.421/kg).

For field work, it is necessary to purchase diesel fuel for EUR 100.44 thousand. The cost of 1 kg of rapeseed-oil-based biodiesel is EUR 0.308. The costs of producing the required amount of biodiesel are presented in Table 8. The amount of expenses, taking into account the annual production program, amounted to EUR 77.727 thousand. The economic effect will be EUR 26.803 thousand. The payback of this event is 1.36 year.

**Table 8.** Costs of biodiesel for the entire sown area of the enterprise, EUR thousand.

| Crop | September | April | May | June | July | August | September |
|---|---|---|---|---|---|---|---|
| Winter wheat | 0.00 | 4.14 | 1.27 | 1.35 | 0.17 | 3.51 | 1.58 |
| Spring wheat | 0.00 | 0.77 | 0.95 | 0.00 | 0.12 | 0.00 | 1.01 |
| Barley | 0.00 | 1.33 | 1.63 | 0.00 | 0.20 | 0.00 | 1.75 |
| Oats | 0.00 | 0.74 | 0.91 | 0.00 | 0.11 | 0.00 | 0.70 |
| Sunflower | 0.57 | 3.26 | 0.56 | 0.74 | 0.46 | 0.11 | 0.00 |
| Perennial grasses | 0.00 | 0.00 | 0.24 | 0.00 | 4.54 | 0.00 | 0.00 |
| Annual grasses | 2.16 | 0.00 | 1.77 | 0.00 | 0.00 | 5.21 | 0.00 |
| Silage Corn | 4.80 | 0.00 | 0.60 | 2.80 | 0.67 | 0.51 | 0.00 |
| Rape | 0.93 | 2.64 | 0.68 | 0.97 | 0.14 | 0.00 | 0.82 |
| TOTAL | 8.46 | 12.89 | 8.61 | 5.85 | 6.42 | 9.34 | 5.86 |

Table 9 shows a list of technological operations for each of the studied technologies for the cultivation of spring rapeseed. Cultivation technologies differed, first of all, by different intensity of soil cultivation: plowing by 25–27 cm, loosening by 10–12 cm, without mechanical tillage (direct sowing). Each variant of soil cultivation was studied at two levels of fertilization: without the use of fertilizers and against the background of the application of fertilizers at a dose of $N_{81}P_{38}K_{38}$, based on the calculation of the planned yield level of 15 cwt/ha.

**Table 9.** List of technological operations in the studied technologies of cultivation of spring rape.

| Rapeseed Cultivation Technologies Based on: | | |
|---|---|---|
| **Plowing** | **Shallow Loosening** | **"Zero" Tillage and Direct Seeding** |
| 1. Peeling 4–6 cm after harvesting the predecessor | 1. Peeling 4–6 cm after harvesting the predecessor | Without autumn tillage |
| 2. Application of complex mineral fertilizers randomly 1.5 c/ha diammofoska | 2. Application of complex mineral fertilizers randomly 1.5 c/ha diammofoska | |
| 3. Plowing by 25–27 cm with regrowth of weeds and fall of the predecessor | 3. Loosening by 10–12 cm during the growth of weeds and fall of the predecessor | 1. Application of the herbicide Hurricane 2 L/ha during the regrowth of perennial weeds and fall of the predecessor |
| 4. Spring harrowing | 4. Spring harrowing | Without spring tillage |
| 5. Presowing cultivation by 3–4 cm | 5. Presowing cultivation by 3–4 cm | |
| 6. Sowing with a SZ-5.4 seeder with the simultaneous introduction of 0.6 c/ha of ammonium nitrate | 6. Sowing with a SZ-5.4 seeder with the simultaneous introduction of 0.6 c/ha of ammonium nitrate | 2. Sowing with the Amazone DMC seeder with the simultaneous introduction of 1.5 c/ha of Diammofoska and 0.6 c/ha of ammonium nitrate |
| 7. Rolling after sowing | 7. Rolling after sowing | 3. Rolling after sowing |
| 8. Spraying with tank mix herbicide + insecticide + biostimulator | 8. Spraying with tank mix herbicide + insecticide + biostimulator | 4. Spraying with tank mix herbicide + insecticide + biostimulator |
| 9. The introduction of ammonium sulfate by scattering into the feed 2.2 cwt/ha | 9. The introduction of ammonium sulfate by scattering into the feed 2.2 cwt/ha | 5. The introduction of ammonium sulfate by scattering into the feed 2.2 cwt/ha |
| 10. Spraying with a tank mixture insecticide + biostimulator | 10. Spraying with a tank mixture insecticide + biostimulator | 6. Spraying with a tank mixture insecticide + biostimulator |
| 11. Mowing into rolls | 11. Mowing into rolls | 7. Mowing into rolls |
| 12. Selection and threshing of rolls | 12. Selection and threshing of rolls | 8. Selection and threshing of rolls |
| 13. Oilseed transportation | 13. Oilseed transportation | 9. Oilseed transportation |
| 14. Primary cleaning of oilseeds | 14. Primary cleaning of oilseeds | 10. Primary cleaning of oilseeds |

Thus, six variants of technologies for cultivation of spring rapeseed, differing in the levels of costs for their implementation, have been put forward for study (Table 10). The use of technologies based on minimum tillage and direct sowing can reduce the cost of fuels and lubricants, on average, more than twofold, from 50% to 26% in the share of total cost. However, with direct sowing, the need for a double increase in the use of crop protection products increases: from 5.7% with traditional technology to 18.5% of the total cost with direct sowing.

**Table 10.** Comparative economic efficiency of technologies for cultivation of spring rapeseed.

| Indicators | Using Plowing | | Shallow Moldless Processing | | Direct Seeding | |
|---|---|---|---|---|---|---|
| | Without the Use of Fertilizers | $N_{81}P_{38}K_{38}$ | Without the Use of Fertilizers | $N_{81}P_{38}K_{38}$ | Without the Use of Fertilizers | $N_{81}P_{38}K_{38}$ |
| Seeds, EUR/ha | 11.76 | 11.76 | 11.76 | 11.76 | 11.76 | 11.76 |
| Fertilizers, EUR/ha | 0.00 | 36.41 | 0.00 | 35.61 | 0.00 | 36.41 |
| Plant protection products, EUR/ha | 5.59 | 5.59 | 5.89 | 5.89 | 12.25 | 12.25 |
| Fuels and lubricants, EUR/ha | 48.53 | 49.67 | 32.39 | 33.52 | 17.39 | 17.95 |
| Repair of equipment, EUR/ha | 2.05 | 2.09 | 1.85 | 1.91 | 1.74 | 1.78 |
| Road transport, EUR/ha | 0.32 | 0.44 | 0.23 | 0.41 | 0.20 | 0.48 |
| Electricity, EUR/ha | 0.16 | 0.23 | 0.13 | 0.22 | 0.10 | 0.25 |
| Wages, EUR/ha | 8.44 | 8.80 | 7.69 | 8.07 | 5.20 | 5.59 |
| Total: variable costs, EUR/ha | 76.85 | 114.99 | 59.93 | 97.39 | 48.65 | 86.48 |
| Depreciation deductions, EUR/ha | 20.48 | 20.92 | 18.52 | 19.03 | 17.35 | 17.83 |
| Total: fixed costs, EUR/ha | 20.48 | 20.92 | 18.52 | 19.03 | 17.35 | 17.83 |
| Total cost, EUR/ha | 97.33 | 135.91 | 78.45 | 116.42 | 66.00 | 104.31 |
| Total revenue, EUR/ha | 137.93 | 168.82 | 92.78 | 144.94 | 70.73 | 153.88 |
| Profit, EUR/ha | 40.60 | 32.91 | 14.33 | 22.95 | 3.90 | 44.16 |
| Profitability, % | 41.72 | 24.21 | 18.27 | 19.72 | 5.91 | 42.34 |
| Cost of 1 ton, EUR | 59.40 | 68.30 | 78.88 | 63.80 | 80.07 | 61.35 |

Increasing the yield of rapeseed is a fundamental factor in obtaining gross harvest and sales proceeds. The high cost of fertilizers and, as a result, a large share of investments in the sector of variable costs, is reflected in the profitability of the production of spring rapeseed with various cultivation technologies [60–63].

In the developed technology of direct sowing with the use of a full range of fertilizers, despite the high level of operating costs, the highest profitability is achieved—42.34%. This is primarily due to an increase in the yield of oilseeds of rape: from 16.4 cwt/ha for plowing to 17.0 cwt/ha for "zero" tillage and direct sowing due to the moisture retained by surface plant residues in the soil and full provision of plant nutrition due to the applied mineral fertilizers. The use of direct sowing technology can reduce production costs by more than EUR 0.032 thousand/ha. This is due to a decrease in the cost of performing energy-intensive tillage operations—deep plowing and subsequent presowing tillage [64–66]. Direct sowing technology allows one to reduce fuel consumption per hectare by half compared to the technology taken for control (using plowing) from 55 to 23.6 kg/ha and reduce labor costs per unit of production to 0.87 man-h/t produced oilseeds.

Thus, on the basis of the performed calculations of economic efficiency in the production of spring rape oilseeds, along with the traditional technology in the Samara region, it is advisable to use the technology of direct sowing with the introduction of a full range of fertilizers. This technology under the conditions of the growing season, i.e., insufficient moisture supply, allows one to obtain a higher yield and reduces production costs per unit of manufactured product—rapeseed oil. The use of direct sowing technology can reduce the cost of fuels and lubricants, on average, more than twofold, compared to conventional

technology with plowing, and reduce the per hectare consumption of motor fuel and labor costs by reducing energy-intensive operations [67–70].

We identified agricultural crops cultivated in the Samara region and suitable for the manufacture of oil (Table 11). In this regard, the interest in considering the economics of the cultivation of these oilseeds is growing, including studying the features of their place in crop rotation and technology, and the influence of these factors on the cost. The final value of the cost of each oilseed crop will be the minimum value among the considered cultivation technologies.

**Table 11.** Economic efficiency of oilseeds cultivation.

| Indicators | Rape | Sunflower | False flax | Mustard | Pumpkin | Flax | Soy |
|---|---|---|---|---|---|---|---|
| Seeds, EUR/ha | 11.76 | 6.84 | 0.82 | 16.36 | 2.25 | 85.23 | 42.95 |
| Fertilizers, EUR/ha | 36.41 | 26.70 | 7.43 | 2.67 | 1.74 | 8.55 | 30.51 |
| Plant protection products, EUR/ha | 12.25 | 2.80 | 9.45 | 4.42 | 4.91 | 5.88 | 10.80 |
| Fuels and lubricants, EUR/ha | 17.95 | 21.52 | 16.48 | 21.93 | 15.00 | 36.76 | 38.98 |
| Repair of equipment, EUR/ha | 1.78 | 1.45 | 1.63 | 1.86 | 1.31 | 1.98 | 2.24 |
| Road transport, EUR/ha | 0.48 | 0.40 | 0.17 | 0.42 | 0.20 | 0.40 | 0.51 |
| Electricity, EUR/ha | 0.25 | 0.20 | 0.08 | 0.17 | 0.07 | 0.22 | 0.28 |
| Wages, EUR/ha | 5.59 | 8.66 | 4.77 | 8.57 | 10.68 | 8.57 | 8.80 |
| Total: variable costs, EUR/ha | 86.48 | 68.58 | 40.83 | 56.41 | 36.16 | 147.57 | 135.07 |
| Depreciation deductions, EUR/ha | 17.83 | 18.64 | 16.61 | 20.98 | 15.08 | 20.40 | 21.48 |
| Total: fixed costs, EUR/ha | 17.83 | 18.64 | 16.61 | 20.98 | 15.08 | 20.40 | 21.48 |
| Total cost, EUR/ha | 104.31 | 87.22 | 57.44 | 77.39 | 51.24 | 167.97 | 156.55 |
| Total revenue, EUR/ha | 160.76 | 123.56 | 74.68 | 102.25 | 62.22 | 226.01 | 208.69 |
| Profit, EUR/ha | 56.45 | 36.34 | 17.24 | 24.86 | 10.98 | 58.05 | 52.15 |
| Profitability, % | 54.1 | 41.7 | 30.0 | 32.1 | 21.4 | 34.6 | 33.3 |
| Cost of 1 ton, EUR | 61.15 | 124.13 | 92.36 | 70.45 | 136.41 | 103.10 | 78.83 |

An example of revealing the dependence of the cost of cultivating oilseeds on the technology of their cultivation is provided for spring rape. Spring rapeseed was selected as a sample crop taking into account the highest oil yield (1000 kg of oil per hectare). Further crops are presented in descending order of the value of this indicator.

In the Samara region, a certain technology for growing sunflower has developed, which allows one to obtain high yields at an earlier date while reducing financial and labor costs. Taking into account the use of new hybrids and varieties with improved characteristics, farmers are able to achieve excellent results in this important branch of agriculture.

Growing sunflower in accordance with the most promising technology allows one to receive a good income from this industry, since sowing requires about 10 kg of seeds per ha, and the yield per hectare can reach 25 cwt. Moreover, not only vegetable oil is obtained from the collected seeds, but also meal, husk and cake, which can become a tangible additional source of income.

Compliance with crop rotation when growing sunflower with the correct alternation of crops in the field is the key to a successful harvest. Sunflower seeds can be sown in the same area no earlier than 6 years after the last sunflower crop, otherwise the seeds of broomrape and pathogens will accumulate in the ground, which can have an extremely negative effect on the crop.

Mineral and organic fertilizers applied in sufficient quantities contribute to the increase in yield and acceleration of the development of the sunflower crop. Throughout the growing season, the sunflower crop needs phosphorus, nitrogen, potassium fertilizers, as well as trace elements such as boron, zinc and manganese.

False flax is a small-seeded oilseed crop of the cruciferous family. It is an annual plant with an erect, branched stem up to 100 cm tall. Agricultural enterprises are interested in obtaining oil on unproductive lands, which will strengthen the economy of the economy (average yield of 10–13 cwt/ha). Its seeds contain over 40% oil and 30% crude protein.

The most promising technology is the cultivation of spring camelina using herbicidal fallow, the preparation of which is carried out using soil-protective moisture-saving technology [71].

Mustard is a triple-industry crop due to its widespread use. It is grown for high quality edible oil, mustard powder and green animal feed. In addition, mustard is widely used as green manure crops, since it has the unique property of absorbing difficult-to-reach forms of nutrients from the soil and transforming them into easily digestible forms.

The average pumpkin yield is relatively low—13.7 cwt/ha. On average, the mass of 1000 seeds in pumpkins does not exceed 420 g. The seeds remain viable for up to 6–8 years. Pumpkin seeds are most responsive to the application of manure or other organic fertilizers in doses of 20–40 t/ha. This procedure is performed after the main plowing, presowing cultivation or cutting of irrigation furrows. With intensive technology, mineral fertilizers are applied simultaneously with sowing. The oil content in oil flax seeds reaches 32–48%. In seven-to-eight-field crop rotations, flax should occupy no more than one field, and return it to its original place no earlier than 6–7 years later. In crop rotations, flax is placed after the best predecessors: perennial grasses, according to the seam turnover, cereals, legumes, winter crops, sown in pairs or after grasses, as well as after row crops. Soybeans are the most widespread pulses of the world. Its seeds contain on average 36–42% of complete protein, consisting of globulins and a small amount of albumin, 19–22% of semi-drying oil and up to 30% of carbohydrates. The best predecessors of soybeans in many areas of its cultivation include green manure fallows, the layer and turnover of the layer of perennial grasses, cereals and spike crops that grow in clean and busy pairs, as well as row crops (corn, potatoes, sugar beets, sweet potatoes, etc.) Soybeans are not grown after Sudanese grass and sunflower; it is also not recommended to sow it after corn, to which the herbicides simazine and atrazine were applied. The optimal saturation of crop rotations with soy is from 22 to 40%. To avoid the negative impact of repeated crops on the yield of soybeans, it is recommended to return it to the field no earlier than after 4 years. In the calculation, we neglect such crops as corn, oats and lupine due to the low content of seed oil (less than 200 kg of oil per hectare). This indicator is twofold lower in comparison with the oilseeds described above. Thus, we assume a decrease in the oil yield in these crops.

The results of the financial analysis of crop producers of the region show that no more than 15% of financially stable agricultural organizations producing oil crop receive government support for fuel and lubricant materials. The analysis of crop producers of the region and social survey of managers and experts reveals that their situation is compounded by price disparities, lack of technical equipment, high costs of fuel, and lack of outlets for crop products. This has identified the need for the development of an economic mathematical model that allows one to calculate the optimal level of government support for every type of biofuel considering forms of government support [72–77].

The developed methodology allows for fast calculation of the optimal level of government support for biofuel production based on three scenarios (for all forms of business organization) using the Microsoft Excel tool [78–80]. We developed a model for the optimization of government support for the production of biofuel by agricultural enterprises growing oil crops, considering the directions of financing [81].

$X_{1-8}$—types of oil crops;

$X_{9-14}$—main directions of government support;

$X_{15-17}$—forms of business organization

The model uses the following notations:

Z—the level of state support for the cultivation of the i-th crop for biofuel production;

$C_i$—costs of the unit for the i-th production technology;

$X_i$—lookup value of the i-th variable denoting the production technology and the level of support;

$A_i$—market output per unit for the i-th production technology;

$a_{ij}$—availability of the j-th resource per i-th unit;

$b_{ij}$—the amount of cash per year;

$N_i$—minimum area under the i-th oil crop used for the production of biofuel, cultivation of which must be guaranteed;

$K_i$—limited i-th production;

$Q$—limitations of economic resources

Objective functions:

optimization of the level of government support considering the main types of oil crops used for biofuel production to provide the region with biofuel:

$$Z = \sum C_i * X_i \rightarrow \max; \tag{4}$$

maximization of the area under oil crops used for the production of biofuel to increase the profit:

$$A = \sum A_i * X_i \rightarrow \max; \tag{5}$$

minimization of costs while achieving the production volume that provides the households with enough biofuel:

$$C = N_i \sum a_{ij} * X_i \rightarrow \min. \tag{6}$$

The model allows us to obtain an optimal structure of production volume, operating costs, gross fuel price and fuel commodity price, as well as the expected profit in the context of a form of business organization [82,83]. Note that this model considers the optimal distribution of government support for all forms of business organizations (costs).

The main objective of the developed methodology is to achieve the optimal production volume at minimal cost, which would allow for the determination of the optimal level of government support for biofuel production to provide per household. A solution to this problem would allow the development of an optimal and effective system for the government regulation of biofuel production.

In accordance with the objective functions, we consider three scenarios of optimal government support:

1. the first scenario allows for optimization of the level of government support considering the levels of agricultural production for the i-th crop to provide farms of the region;
2. the second scenario allows for the determination of the maximum profit from the biofuel production through increased agriculturally used areas;
3. the third scenario allows for the calculation of the minimum expenses of achieving the volume of production that provides the farm with raw materials.

According to our calculations based on the first scenario, the optimum level of government support for the field should be EUR 13.223 million. In this case, financial resources should be distributed to the following targets:

- preferential tax, loan and financing systems for producers accumulate funds of EUR 5.29 million;
- development of biofuel production—EUR 4.63 million;
- crop insurance—EUR 1.98 million;
- development of information consulting service and technical equipment—EUR 1.044 million;
- other targets—EUR 0.28 million.

In the implementation of the second scenario in the Samara region, the agriculturally used area planted with oil crops should be increased by 47.1 thousand ha. This would solve the problem of providing farms producing biofuel with raw materials, and improve the difficult financial situation of some producers.

In accordance with the third scenario, budgetary expenditures for the achievement of planned standards of production will equal EUR 370.08 thousand. The difference between the gross cost and diesel fuel per farm is EUR 0.98 thousand.

The most beneficial for agricultural organizations would be the reduction in tax payments to the federal and regional budgets.

## 4. Conclusions

As a result of the present study, it was determined that the production of homemade biodiesel under the conditions of an agricultural enterprise is economically justified, even at low global oil prices. The values are determined from the calculation of the most successful cultivation technologies for each crop. The most profitable oilseeds for cultivation are rapeseed and sunflower, it is interesting that these crops are widely used in the production of biofuel. The amount of diesel fuel used to cultivate the planted area of the Samara region is 77,062.9 tons of diesel fuel. Based on described calculations of comparative effectiveness of biofuel production using MIXER-2 11 AB with a production capacity of 5000 tons per year, we were able to determine that 15 units will be enough to provide agricultural producers of the Samara region with biofuel. For greater convenience, 5 units should be located in every agricultural zone of the Samara region.

In the course of the work, a business plan was developed for an investment project to create a mini-biodiesel production workshop. The required investment amount is EUR 29.2 thousand, which consists of the acquisition of equipment and the reconstruction of a grain warehouse in the farm. Over the 5 years of the project, a net discounted income of EUR 0.032 million will be obtained, and the payback period is 25 months at a discount rate of 11%.

Practical implementation of the proposed mathematical model includes the involvement of leading experts in the field of cultivation of oil crops, who can describe these functions on a quantitative basis. There is also the need for extensive testing of the model and its identification capacity, based on insufficient and inaccurate data, to estimate parameters and correct the expert functions. Since we need to develop a different system of government regulation of this field based on modern information technologies, we used methods of mathematic modeling.

The developed methodology allows us to develop an optimal structure of biofuel production, estimate money and labor costs, gross and commodity value of biofuel, and expected margin of all organizational forms of production. Generally speaking, this will allow for the optimization of tools in terms of the effective use of budget funds and the provision of the Samara region with biofuel.

It is essential to point out that issues related to biofuel production are not only related to government support. Measures of government support can only be effective if they are based on the identification and implementation of internal growth reserves, and the effective production of biofuel, which will result in excellent performance of farms that cultivate oil crops and produce biofuel and the agrifood complex. Growing crops for biodiesel production instead of food crops does not lead to a food security problem. Currently, the region produces much more grain than is required for domestic consumption and consumption in neighboring regions. A lot of grain is exported abroad, while dealers receive profit from this. At the same time, according to the requirements of crop rotation, the list of crops that can be grown instead of grain is limited. And the use of spring rapeseed for biodiesel production can partially reduce the supply of grain in the region and, as a result, increase profitability for agricultural enterprises.

**Author Contributions:** Conceptualization, K.A.Z., L.N.Z. and V.V.N.; methodology, O.K.K.; software, V.V.N. and O.K.K.; validation, T.V.M., V.P.A. and V.G.P.; formal analysis, Y.D.S., G.V.K. and A.A.T.; data curation, K.A.Z., E.A.K. and R.R.G.; writing—original draft preparation, O.K.K. and K.A.Z.; writing—review and editing, V.V.N.; supervision, V.P.A.; project administration, L.N.Z. All authors participated in collecting and identifying files, drafting the manuscript. All authors participated in the study design and coordination and interpreted the data. All authors have read and agreed to the published version of the manuscript.

**Funding:** The authors (V. Nosov, V. Plyushchikov, V. Avdotin) took part in the article with the support by the RUDN University Strategic Academic Leadership Program.

**Institutional Review Board Statement:** Ethical review and approval were waived for this study not applicable.

**Informed Consent Statement:** Not applicable.

**Data Availability Statement:** Data supporting the conclusions of this article are presented in the main manuscript.

**Conflicts of Interest:** The authors declare that they have no competing interest.

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
