# Peer review of "Agronomic and Economic Aspects of Biodiesel Production from Oilseeds: A Case Study in Russia, Middle Volga Region"

_agriculture, doi:10.3390/agriculture12101734_

Round 1

Reviewer 1 Report

Abstract should contain numeral values that depict the fitness the models

In the Introduction part, a region map depiction is required for easy understanding of readers.

In M&M, do mention specifically about the the software and their input entries. Showing jusy window is insufficient to take by other researchers.

Is it developed by the same researchers? If not, do mention the license no. etc.,

Why authors want to focus on spring rape? Justify.

DO refer line no. 143; crops with minimum concentrations of polyunsaturated fatty acids, such as linoleic 144 acid (18:3);

The premise is incorrect. With the increase of more unsaturation the quality of the fuel is poor due to oxidation and low cetane index. Authors are suggested to write correct propositions.

Cultivation of oilseeds in place of food crops doesn't leads to food vs fuel dilemma in the region? Justify.

Cost of the biodiesel process from upstream to downstream must be taken for consideration?

Are these analyses were backed by an statistical software? If yes, then auuthors are encouraged to give P-value and R square value.

The intro part and conclusion part should emphasize the significance of the work.

The paper may be considered for publication, if the authors have addressed the paper corrections.

Author Response

Thanks for the good review.

Reviewer 2 Report

Agriculture
Agronomic and economic aspects of biodiesel production from 2 oilseeds: a case study in Russia, Middle Volga region

The manuscript is of general interest. The following comments should help further improve the quality of the work:

1-Abstract should be improved by including the major findings of the work quantitatively.
2-Please add one more keyword (up to 6 is allowed). Metadata including keywords are important in terms of the searchability of the manuscript if published.
3-The Keywords “ state support; modeling; crops; technology “ are not suitable and should be replaced with more appropriate ones.
4-The novelty/originality of the paper should be more effectively established.
5-Reference lumping should be avoided. Please cite references where they exactly belong; this will prevent reference lumping.
6-Too short paragraphs should be avoided.
7-Please check the manuscript for the use of punctuations.
8-“biogenic energy carriers” should be replaced with “geogenic energy carriers”.
9-Biodiesel is an important part of this work and hence, latest trends in biodiesel production and processing should be briefly discussed by referring to the recently published review/research works, including “Machine learning technology in biodiesel research: A review”, Qualitative role of heterogeneous catalysts in biodiesel production from Jatropha curcas oil”, etc.
10-There are some wrong technical information in the manuscript, such as the following:
“low a number of components content in exhaust gases, such as …, nitrogen oxides NOX …”. Biodiesel does not lead to less NOX in general and in fact, it is the opposite. Most studies report increased NOX emissions attributable to the high oxygen content of biodiesel. Please ensure of the technical accuracy of the statements made in the manuscript.
11-All Equations should be properly numbered and referred to (mentioned) in the text.
12-Please make sure all the units will be presented in compliance with the SI System. For instance, please use "kg" for "Kilogram", “L” for “Liter”, etc. This comment applied to the units used in Figures/Tables too.
13-Table 2 is not necessary and could be presented in the text. Please remove it.
14-Table 2: the symbol used for degree centigrade is wrong; it is like zero “0” has been used. Please fix this issue.
15-Please remove the row “including:“
16-The structure of Table 9 is not suitable and should be improved.
17-Equations 4, 5, and 6 have been presented as figures and are blurred and not appropriate. Please write the equations in the MS Word instead of inserting them as figures.
18-Overall, the organization of the manuscript should be substantially improved.
19-Please change "4. Conclusions" to "4. Conclusions and prospects". Accordingly, please elaborate on the future research needs in this domain.
20-Future efforts should look into the sustainability features of the results obtained using advanced sustainability assessment tools, including life cycle assessment, exergy, etc. In a recent work “The role of sustainability assessment tools in realizing bioenergy and bioproduct systems”, Aghbashlo and colleagues have elaborated on the significance of these methods. Please refer to this article and discuss this as a future research need.
21-More comparison of the results obtained with those of previous studies and critical discussion should also be added.
22-The practical implications of the present study should be discussed as well.
23-Please also discuss the limitations of the present study.

Author Response

Dear reviewer! Thank you for your feedback on our article.

Reviewer 3 Report

The manuscript is written well and I do not find any improvement required in the manuscript. It can be accepted for publication just by checking the grammar throughout the manuscript. 

Author Response

Dear reviewer!
Thank you for appreciating our work. English spelling will be corrected.